# TRACE: ADAPTIVE CURTAILMENT OF REASONING IN RETRIEVAL-AUGMENTED GENERATION VIA TRAJECTORY REFLECTION

## ABSTRACT

Large Reasoning Language Models (LRLMs) excel at complex reasoning tasks by generating multi-step chains of thought. However, their autoregressive nature can lead to overthinking, a tendency to generate overly verbose reasoning that inflates computational costs and can even degrade accuracy. Advanced methods mitigate overthinking by monitoring the model's internal confidence and terminating the process once a high-confidence answer is found. This strategy is effective when models reason using their parametric knowledge, but it faces significant challenges and risks failure in Retrieval-Augmented Generation (RAG) scenarios where external knowledge is introduced. We have conducted an in-depth analysis of this issue and reveal that the reasoning process in RAG universally follows a distinct, two-stage **Exploratory-Synthesizing** pattern. Unlike scenarios that rely solely on parametric knowledge where confidence gradually accumulates, the initial exploration phase in this pattern involving external documents exhibits **premature confidence**, where models become highly certain after inspecting only partial evidence. This early confidence surge misleads conventional termination methods, causing them to halt the process prematurely and produce incorrect answers. To address this, we propose Trajectory Reflection with Adaptive Curtailment and Exit (**TRACE**), a training-free framework that employs a cascading check at each reasoning step. First, it monitors the stability of the model's predictive beliefs to ensure sufficient knowledge exploration. Subsequently, it assesses task completion by confirming high confidence in a synthesized final answer. Extensive experiments demonstrate that TRACE reduces token generation by 22% to 54% while achieving comparable or superior accuracy to standard Chain-of-Thought prompting.

## 1 INTRODUCTION

Large Reasoning Language Models (LRLMs) have become a dominant paradigm in AI research, largely due to their ability to emulate human-like Chain-of-Thought (CoT) processes for solving complex problems (Wei et al., 2022). This capability has led to strong performance on tasks like scientific reasoning and multi-hop question answering (Yang et al., 2018; Kojima et al., 2022). However, this progress faces a significant efficiency challenge known as "overthinking", where models generate overly verbose reasoning steps (Wu et al., 2025; Wei et al., 2025). This increases computational costs and latency, hindering practical deployment and sometimes even degrading accuracy by introducing irrelevant information.

Various strategies have been developed to improve reasoning efficiency. Early methods focused on post-hoc optimization, which involves refining a completed reasoning chain by pruning or compressing it (Fu et al., 2024). Others have used prompt-based guidance, designing specific prompts to steer models toward more concise outputs (Chen et al., 2023; Sui et al., 2024). More recently, research has shifted toward dynamic, output-based methods that intervene during the inference process itself. These methods often monitor intermediate answer consistency (Wang et al., 2022) or token-level generation confidence (Zhang et al., 2024). A state-of-the-art example is DEER (Yang et al., 2025), a confidence-based early exit strategy that terminates reasoning once the model's confidence in an answer exceeds a set threshold. Despite their success, these frameworks are built on the assump-

tion that reasoning depends solely on the model's internal, parametric knowledge. This assumption conflicts with the principles of RAG (Lewis et al., 2020), which requires models to synthesize information from multiple and complex documents (Zhao et al., 2024). This discrepancy raises a crucial question regarding the reliability of confidence-based heuristics.

This paper investigates this question and demonstrates that external documents introduce a reasoning pattern that undermines existing efficiency solutions. Our analysis reveals that in RAG settings, models often become highly confident after processing only partial evidence, a behavior that misleads termination heuristics like DEER. We identify this process as a two-stage "Exploratory-Synthesizin" paradigm. To address this challenge, we introduce TRACE (Trajectory Reflection with Adaptive Curtailment and Exit), a novel framework that actively monitors the reasoning process. TRACE uses a cascading check at each step: it first ensures sufficient exploration by confirming the stability of the model's belief state, and only then does it verify high confidence in a synthesized conclusion before terminating.

The primary contributions of this work are threefold:

- **We systematically analysis and validate the failure of confidence-based methods in RAG.** We demonstrate why confidence-based early exit strategies are unreliable in multi-source RAG scenarios. We identify a unique "Exploratory-Synthesizing" pattern that leads to premature model confidence, providing quantitative evidence that this failure mode causes accuracy degradations of up to 5.74 percentage points.

- **We propose TRACE, a novel training-free framework designed for robust reasoning in RAG.** TRACE introduces a dual-check mechanism that addresses the challenge of premature confidence. It first ensures sufficient knowledge exploration by monitoring the stability of the model's belief state, and only then uses high answer confidence as a signal for termination. Our ablation studies confirm that both checks are critical to its success.

- **We demonstrate the effectiveness and efficiency of TRACE with extensive experimental evidence.** Across four benchmarks and five different LRLMs, TRACE achieves accuracy comparable to or even better than a standard Chain-of-Thought baseline while being substantially more efficient. Specifically, it reduces the number of generated tokens by 22% to 54%, successfully mitigating the "overthinking" problem in RAG scenarios.

## 2 RELATED WORK

Recent research on LRLMs has focused on improving reasoning efficiency to address "overthinking" a phenomenon where models generate excessively long Chains-of-Thought that increase inference latency (Sui et al., 2024; Wang et al., 2025). Following established taxonomies (Sui et al., 2024; Kumar et al., 2024), existing solutions can be broadly divided into three categories: post-training, prompt-based, and output-based methods.

**Post-Training Methods.** This approach modifies a model's internal parameters through additional training. Techniques range from supervised fine-tuning and reinforcement learning, which teach models to generate adaptive-length CoTs (Fu et al., 2024), to using latent representations to replace explicit textual reasoning (Sui et al., 2024). However, these methods are often computationally expensive, require complex dataset construction, and risk overfitting, which can harm generalization.

**Prompt-Based Methods.** This category focuses on guiding model output without altering its weights. These methods use carefully engineered prompts, sometimes adapted to query difficulty, to encourage more concise reasoning (Zhou et al., 2022). While somewhat effective, prompt engineering alone often lacks the fine-grained control needed to manage the step-by-step process of overthinking.

**Output-Based Methods.** This approach dynamically controls the generation process by intervening during inference. Our proposed framework, TRACE, is a novel method in this category. Other output-based techniques also aim for early termination but often have limitations. For instance, some require an auxiliary "probe" model to verify intermediate answers, adding architectural complexity

(Kadavath et al., 2022; Kumar & Sarawagi, 2019). Others are only optimal in specific scenarios, such as low-budget regimes or with best-of-N sampling (Wang et al., 2022).

## 3 MOTIVATION AND OBSERVATION

### 3.1 FAILURE OF CONFIDENCE HEURISTICS UNDER RAG

Confidence-based early exit strategies like DEER (Yang et al., 2025) have proven effective for reducing inference costs in scenarios where a model relies on its internal parametric knowledge. To evaluate their robustness in RAG, we conducted paired experiments comparing DEER against a vanilla CoT baseline. The results are summarized in Table 1.

In the non-RAG condition, DEER preserves task accuracy relative to the CoT baseline, with minor differences consistent with prior findings (Yang et al., 2025). By contrast, the RAG condition reveals a systematic and substantial degradation in performance. Accuracy declines are an order of magnitude larger, with observed drops ranging from 2.67 to 5.74 percentage points across all tested models and datasets. The magnitude and consistency of this decline indicate that model-reported confidence becomes a brittle and misleading indicator of global synthesis readiness when reasoning requires the integration of multiple external documents.

Table 1: **Comparison of accuracy for the DEER method relative to the Vanilla CoT baseline in RAG and non-RAG (w/o RAG) settings.** 'Diff' represents the absolute accuracy difference (DEER Acc. - Vanilla CoT Acc.). Negative values indicate a performance drop for DEER.

| Method | NQ | | TriviaQA | | SQuAD | | HotpotQA | |
|---|---|---|---|---|---|---|---|---|
| | RAG | w/o RAG | RAG | w/o RAG | RAG | w/o RAG | RAG | w/o RAG |
| ***Qwen3-8B*** | | | | | | | | |
| Vanilla CoT | 68.95 | 46.71 | 87.72 | 78.65 | 68.59 | 35.60 | 55.17 | 39.00 |
| DEER | 63.52 | 46.47 | 84.37 | 78.29 | 65.45 | 35.41 | 51.57 | 38.75 |
| **Diff** | **-5.43** | **-0.24** | **-3.35** | **-0.36** | **-3.14** | **-0.19** | **-3.60** | **-0.25** |
| ***Qwen3-14B*** | | | | | | | | |
| Vanilla CoT | 68.89 | 51.46 | 87.93 | 84.43 | 69.02 | 39.52 | 56.52 | 43.70 |
| DEER | 65.17 | 51.29 | 85.26 | 84.55 | 64.13 | 39.40 | 51.80 | 43.52 |
| **Diff** | **-3.72** | **-0.17** | **-2.67** | **+0.12** | **-4.89** | **-0.12** | **-4.72** | **-0.18** |
| ***Deepseek-Distill-Qwen-7B*** | | | | | | | | |
| Vanilla CoT | 62.19 | 23.88 | 83.70 | 50.21 | 59.60 | 18.17 | 47.70 | 21.83 |
| DEER | 58.20 | 23.69 | 79.53 | 49.94 | 54.57 | 17.91 | 43.28 | 21.61 |
| **Diff** | **-3.99** | **-0.19** | **-4.17** | **-0.27** | **-5.03** | **-0.26** | **-4.42** | **-0.22** |
| ***Deepseek-Distill-Qwen-14B*** | | | | | | | | |
| Vanilla CoT | 67.84 | 44.50 | 87.68 | 79.28 | 66.56 | 32.77 | 56.09 | 37.50 |
| DEER | 63.93 | 44.63 | 84.35 | 79.24 | 60.82 | 32.58 | 51.95 | 37.39 |
| **Diff** | **-3.91** | **+0.13** | **-3.33** | **-0.04** | **-5.74** | **-0.19** | **-4.14** | **-0.11** |
| ***Llama3-8B*** | | | | | | | | |
| Vanilla CoT | 68.34 | 49.12 | 87.14 | 79.33 | 67.79 | 37.48 | 57.38 | 41.76 |
| DEER | 63.04 | 48.89 | 82.91 | 79.05 | 63.55 | 37.31 | 52.78 | 41.52 |
| **Diff** | **-5.30** | **-0.23** | **-4.23** | **-0.28** | **-4.24** | **-0.17** | **-4.60** | **-0.24** |

### 3.2 PREMATURE CONFIDENCE

To identify the mechanism responsible for this empirical failure, we analyzed the temporal evolution of model confidence during the generation process. To enable a standardized comparison across reasoning trajectories of varying lengths, we first define a "thinking step". In our experiments, a "step" is a segment of reasoning generated by the model until a '\n\n' delimiter is produced, with a maximum length of 150 tokens per step. We use the full trajectory length generated by the Vanilla CoT method as the baseline for normalizing the "Thinking Progress" for all methods. Furthermore, to ensure our analysis focuses on complex problems that require multi-step reasoning, our statistics only include samples where the Vanilla CoT baseline generated more than 10 steps.

On the HotpotQA dataset, these valid samples constitute 87.62% of the total. Figure 1 presents the confidence evolution trajectories under these standardized and filtered conditions.

In the non-RAG condition, the trajectory follows an expected convergent pattern. Confidence begins at a relatively low level and increases steadily as the model accumulates internal deductions and the reasoning process converges on a final answer.

The RAG trajectory, however, departs sharply from this pattern. Here, confidence rises to a substantially higher level during the initial steps and remains elevated even as subsequent steps introduce further evidence. We term this behavior **premature confidence**. This phenomenon stems from a fundamental shift in the role of early reasoning steps within the RAG framework. These initial steps are frequently dominated by the extraction and summarization of retrieved passages, which are operations the model can perform with high local certainty. Consequently, a termination rule may misinterpret this high local certainty as a signal that the global reasoning task is complete. This causes the model to halt prematurely, before it can perform the crucial integrative reasoning required to synthesize a robust final answer from all available evidence. This mismatch between local confidence during evidence extraction and global readiness for synthesis explains the failure mode observed for DEER and related heuristics in RAG settings.

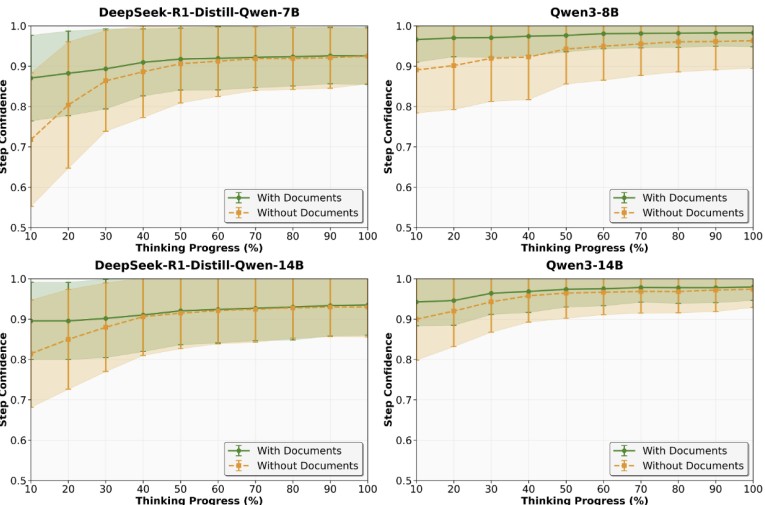

Figure 1: **Comparison of generation confidence for different models on the HotpotQA dataset**, under RAG (green lines, labeled 'with documents') and non-RAG (orange lines, labeled 'without documents') settings. Confidence is quantified by prompting the model for its current answer after each step and calculating the average log-probability of the resulting answer tokens. The solid line represents the mean across the validation set, and the shaded area indicates one standard deviation.

### 3.3 THE EXPLORATORY–SYNTHESIZING PATTERN

The phenomenon of premature confidence reflects a fundamental shift in reasoning dynamics. As illustrated in Figure 2, traditional single-source reasoning that relies on internal knowledge can be conceptualized as a *convergent deduction* process, where successive steps incrementally refine an internal hypothesis until a conclusion is reached (Pan et al., 2024). Retrieval-augmented generation, in contrast, induces a two-stage trajectory that we characterize as the **Exploratory-Synthesizing** pattern.

The process begins with the **exploration** stage, where the model inspects retrieved passages, extracts salient facts, and gathers candidate evidence. The outputs during this phase often exhibit high lexical overlap with the source texts and are consequently associated with high local confidence. This is followed by the **synthesis** stage, during which the model integrates the collected evidence, reconciles any conflicting information, and constructs a coherent evidential chain to support a final answer. A genuine readiness for termination is achieved only during this second stage.

Conventional confidence metrics, however, are unable to distinguish between these two stages. Because they reflect local extraction certainty more strongly than global synthesis completeness, a single confidence threshold conflates the signals from both phases, leading to premature termination. An effective termination policy for RAG must therefore be able to assess both the completeness of the evidence exploration and the stability of the synthesized conclusion. This requirement directly motivates the dual-check mechanism implemented in our TRACE framework (Asai et al., 2023; Yan et al., 2024).

> **Question:** What were the primary factors that contributed to the decline and fall of the Western Roman Empire?
>
> **Document 1:** ...the primary driver was economic. The debasement of the denarius led to runaway inflation, while the tax system became ...
>
> **Document 2:** While economic factors played a role, they were secondary to the military reality. The constant, large-scale migrations of ...

> **Reasoning Without External Documents (non-RAG)**
>
> **Step1:** The fall of the Western Roman Empire was a complex process resulting from multiple interconnected factors ...
>
> **Step2:** Politically, the empire suffered from a century of civil wars, a rapid succession of weak and ineffective emperors, and ...
>
> **Step3:** Economically, factors included hyperinflation caused by the debasement of currency, excessive taxation on the populace ...
>
> **Step4:** Militarily, the empire faced constant external pressure from Germanic tribes and other groups along its borders. Internally, the ...
>
> **...</omitted>**

> **Reasoning With External Documents (RAG)**
>
> **EXPLORATION**
>
> **Step1:** Document 1 argues that the root cause was economic failure, specifically citing inflation, oppressive ...
>
> **Step2:** Document 2 prioritizes military factors, highlighting both external pressure from migrating peoples and the internal ...
>
> **SYNTHESIS**
>
> **Step3:** Therefore, a more comprehensive conclusion based on these sources is that the fall was not due to ...
>
> **...</omitted>**

Figure 2: **Conceptual comparison of the reasoning patterns in non-RAG (left) and RAG (right) scenarios.** The "Convergent deduction" pattern is a linear deductive process. In contrast, the "Exploratory-Synthesizin" pattern in RAG requires an initial Exploration phase to analyze multiple external documents before proceeding to a Synthesis phase to form an evidence-based conclusion.

## 4 METHOD

### 4.1 FRAMEWORK OVERVIEW

To curtail overthinking in RAG scenarios, we introduce TRACE (Trajectory Reflection with Adaptive Curtailment and Exit), a novel reasoning control framework. Unlike conventional approaches where LRLMs generate text uninterruptedly, TRACE operates on an iterative generate-reflect-decide cycle. Following the generation of each reasoning segment, TRACE invokes its core Trajectory Reflection module to determine whether to terminate the process. This module performs a cascading, two-condition check. It first assesses the sufficiency of the model's exploration of external knowledge and subsequently evaluates the completeness of its synthesis into a confident answer. Reasoning is halted only when both conditions are satisfied, thereby balancing thoroughness with computational efficiency. Figure 3 provides a high-level illustration of this framework.

### 4.2 ITERATIVE REASONING TRAJECTORY CONSTRUCTION

We formalize the model's reasoning process as an incremental **Reasoning Trajectory**, $\mathcal{T}_t$, composed of a sequence of **Reasoning Segments** $\{S_1, S_2, \ldots, S_t\}$. In our implementation, each segment $S_t$ represents a single step in the thought process, generated by the model until a '\n\n' delimiter is produced, with a maximum length of 150 tokens per step. At this point, generation is paused to invoke the Trajectory Reflection module. At any step $t$, the LRLM generates a new segment $S_t$ conditioned on the question $Q$, the context $C$, and the preceding trajectory $\mathcal{T}_{t-1}$:

$$S_t \sim P_{\text{LLM}}(S \mid Q, C, \mathcal{T}_{t-1}) \tag{1}$$

After generating $S_t$, the updated trajectory $\mathcal{T}_t = \mathcal{T}_{t-1} \cup \{S_t\}$ is evaluated by the Trajectory Reflection module. The goal of TRACE is to monitor this process and intervene at the optimal moment.

### 4.3 TRAJECTORY REFLECTION

The Trajectory Reflection module is the technical core of TRACE. It quantifies the model's internal cognitive state at each step to decide whether to continue or terminate the reasoning process.

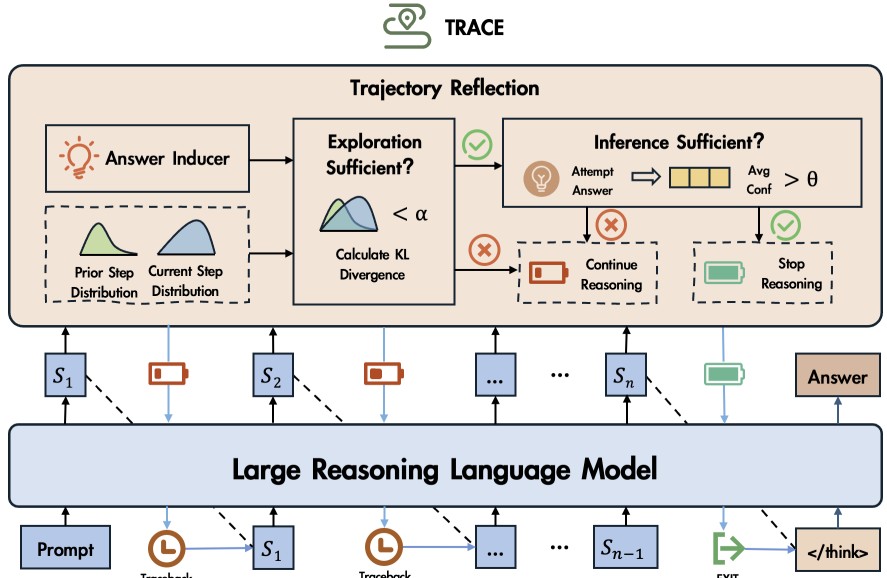

Figure 3: **Overview of the TRACE framework**. At each reasoning step, the LRLM generates a reasoning segment $S_t$. The resulting trajectory state is then passed to the Trajectory Reflection Module for evaluation. This module performs a cascading, two-stage check. First, it assesses whether exploration is sufficient by evaluating the stability of the predictive belief state. Then, Only if this condition is met does the module proceed to assess whether the synthesis is sufficient for a confident answer by checking the confidence score. An EXIT signal is sent to terminate the reasoning process only when both conditions are satisfied. Otherwise, the LRLM is prompted to generate the next segment.

This decision is made through a cascading two-stage check. The complete algorithm is detailed in Algorithm 1.

**Condition 1: Is Knowledge Exploration Sufficient?**    The first check assesses whether the model has adequately explored the external knowledge. Ideally, this would be measured by the "Belief Gain" that a new segment $S_t$ provides over the space of all possible answers, a quantity best captured by KL Divergence (Kuhn et al., 2023). However, computing this over an infinite answer space is intractable.

To efficiently proxy the model's belief state, we introduce a targeted probing method. At each reasoning step $t$, we append the specific instruction (`</think>\boxed{}`) to the current trajectory $T_t$. This instruction is designed to elicit an immediate answer, as our primary prompt directs the model to enclose its final response within a (`\boxed{}`) block. We then capture the model's belief state by using the resulting probability distribution over the entire vocabulary for the first token of this elicited answer, a distribution we denote as $D_t$. It is crucial to clarify that even if the model predicts the same top token at different reasoning steps, $t$ and $t'$, the full probability distributions over the entire vocabulary, $D_t$ and $D_{t'}$, are typically distinct. Although derived from a single token position, this distribution provides a nuanced representation of the model's conviction across all potential answers. The objective of this probing is not to achieve perfect precision, but rather to efficiently identify potential points of reasoning completion. Consequently, while this approximation may introduce minor inaccuracies, it strikes a deliberate balance between computational efficiency and diagnostic effectiveness.

We measure the stability of this belief state by comparing the current distribution $D_t$ with all historical distributions and selecting the minimum KL divergence:

$$\Delta_{KL}^{(t)} = \min_{k\in\{0,...,t-1\}} D_{KL}(D_t||D_k) \tag{2}$$

We use the minimum KL divergence across the entire history for two reasons: 1) it provides a robust measure of global stabilization against minor, local fluctuations between consecutive steps, and 2) it can detect reasoning cycles where the model reverts to a previous belief state. Exploration is considered sufficient when $\Delta_{KL}^{(t)}$ falls below a predefined threshold $\theta_{\text{exp}}$.

**Condition 2: Is Reasoning Sufficient for a Confident Answer?**  Once exploration is deemed sufficient ($\Delta_{KL}^{(t)} < \theta_{\text{exp}}$), the module performs the second check to ensure the model can produce a reliable conclusion. Using the same probing mechanism, we prompt the model to directly generate the answer sequence $W = \{w_1, \ldots, w_N\}$. We then quantify the model's confidence by calculating the geometric mean of the token probabilities:

$$C(W \mid Q, C, \mathcal{T}_t) = \left( \prod_{i=1}^{N} P(w_i \mid Q, C, \mathcal{T}_t, w_{<i}) \right)^{1/N} \tag{3}$$

In line with conventional confidence-based methods, a high confidence score is interpreted as an indication that the model has successfully synthesized the explored knowledge into a coherent conclusion. The reasoning process is terminated if this score exceeds a predefined threshold $\theta_{\text{inf}}$.

**Inference Efficiency.**  The Trajectory Reflection module introduces a minor computational overhead, but this cost is insignificant compared to the substantial efficiency gains from reducing redundant reasoning steps. The use of a KV cache and a short probing prompt means that additional computation is only required for a few new tokens, as the cached representations for the lengthy prior context are reused. A detailed computational cost analysis is provided in Appendix E.

---

**Algorithm 1** The TRACE (Trajectory Reflection with Adaptive Curtailment and Exit) Algorithm

---

1: **Input:** Question $Q$, Context $C$, LRLM $P_{\text{LLM}}$, Max steps $T_{\text{max}}$
2: **Hyperparameters:** Exploration sufficiency threshold $\theta_{\text{exp}}$, Answer confidence threshold $\theta_{\text{inf}}$, Min steps $t_{\text{min}}$
3: **Initialize:** Trajectory $\mathcal{T}_0 \leftarrow \emptyset$, Belief history $\mathcal{H} \leftarrow \{\}$
4: **for** $t = 1, 2, \ldots, T_{\text{max}}$ **do**
5:   *// Generation Phase*
6:   Generate new reasoning segment $S_t \sim P_{\text{LLM}}(S|Q, C, \mathcal{T}_{t-1})$
7:   Update trajectory $\mathcal{T}_t \leftarrow \mathcal{T}_{t-1} \cup \{S_t\}$
8:   *// Trajectory Reflection Phase*
9:   Construct probing prompt and compute current predictive belief distribution $D_t$
10:   **if** $t \geq t_{\text{min}}$ **and** $\mathcal{H}$ is not empty **and** $\min_{D_k \in \mathcal{H}} D_{\text{KL}}(D_t || D_k) < \theta_{\text{exp}}$ **then**
11:     *// Exploration is sufficient; check if reasoning can derive a confident answer.*
12:     Generate candidate answer $W$ and compute its confidence $C(W|Q, C, \mathcal{T}_t)$
13:     **if** $C(W|Q, C, \mathcal{T}_t) > \theta_{\text{inf}}$ **then**
14:       *// Reasoning is sufficient; trigger adaptive exit.*
15:       **break**
16:     **end if**
17:   **end if**
18:   Update belief history $\mathcal{H} \leftarrow \mathcal{H} \cup \{D_t\}$
19: **end for**
20: **Output:** Final answer generated based on the final trajectory $\mathcal{T}_t$.

---

## 5 EXPERIMENTS

### 5.1 EXPERIMENTAL SETUP

**Datasets and Models.**  We evaluate our method on four widely-used benchmarks to cover a range of question types, from open-domain question answering to multi-hop reasoning. The datasets are **Natural Questions (NQ)** (Kwiatkowski et al., 2019), **TriviaQA** (Joshi et al., 2017), **SQUAD** (Rajpurkar et al., 2016) and **HotpotQA** (Yang et al., 2018). To ensure generality, we conduct experiments on five open-source LRLMs: **Qwen3-8B**, **Qwen3-14B**, **Llama3-8B**, **DeepSeek-R1-Distill-Qwen-7B**, and **DeepSeek-R1-Distill-Qwen-14B** (Bai & Qwen Team, 2025; DeepSeek-AI et al.,

2025). All experiments are run using the vLLM inference framework (Kwon et al., 2023). For retrieval, we follow the setup of (Lin et al., 2023), using **ColBERTv2** (Santhanam et al., 2022) as the retriever over the Wikipedia corpus (21M+ passages) from (Karpukhin et al., 2020). All methods share the same retrieved inputs.

**Baselines.** We compare TRACE against four representative baselines: 1) **Vanilla CoT** (Wei et al., 2022; Kojima et al., 2022), a standard baseline where the LRLM generates a Chain-of-Thought over retrieved documents until it terminates naturally; 2) **NoThinking** (Ma et al., 2024), a single-pass, direct-answering method, which we evaluate without its parallel sampling mechanism for a direct comparison of serialized reasoning efficiency; 3) **Dynasor-CoT** (Wang et al., 2022), an adaptive early exit method that terminates when the generated answer remains consistent over several steps; and 4) **DEER** (Yang et al., 2025), a state-of-the-art early exit method that terminates generation once the model's confidence in an answer exceeds a fixed threshold.

**Evaluation Metrics.** We evaluate all methods on two aspects: 1) **Task Performance**, measured by Accuracy, which checks if the correct answer is present in the generated output; and 2) **Reasoning Efficiency**, measured by Average Generated Tokens (Avg. Tokens), which reflects computational cost.

**Hyperparameters.** For TRACE, the core hyperparameters $\theta_{\exp}$ and $\theta_{\inf}$ are set for each dataset via a grid search. We search $\theta_{\exp}$ in $[0.005, 0.015]$ with a step of $0.002$, and $\theta_{\inf}$ in $[0.85, 0.95]$ with a step of $0.02$. Optimal values are determined by rapid validation on a random sample of 100 examples. This process takes approximately 20 minutes on four NVIDIA RTX 5090 GPUs. Meanwhile, we utilize the optimal hyperparameter configurations recommended in the original papers for all baseline methods. The minimum reasoning steps, $t_{\min}$, is fixed at 2 for all experiments. All methods use greedy decoding (temperature = 0.0).

### 5.2 MAIN RESULTS AND ANALYSIS

As shown in Table 2, our TRACE framework consistently achieves a superior balance between task performance and inference efficiency across all models and datasets. Unlike other methods that incur significant accuracy degradation, TRACE maintains accuracy comparable to, and sometimes better than, the Vanilla CoT baseline while significantly reducing the number of generated tokens.

Our analysis highlights a central dilemma in RAG reasoning efficiency, requiring a balance that avoids the pitfalls of both under-thinking and over-thinking. A detailed case study is presented in Appendix A.On one hand, aggressive early-exit strategies like DEER and NoThinking exemplify under-thinking. They terminate on misleading premature confidence signals drawn from partial evidence, resulting in significant accuracy drops. On the other hand, the standard Vanilla CoT baseline is prone to over-thinking. Its unrestrained reasoning chains are not only inefficient due to high token consumption but can also be unstable, as we observe that models may enter a state of "local oscillation" that wastes resources and can degrade performance.

TRACE successfully navigates between these extremes. Its dual-check mechanism first ensures sufficient exploration to prevent premature termination, then prunes redundant steps before overthinking. Compared to Dynasor-CoT, TRACE generally achieves higher accuracy with less token count, which suggests that monitoring the model's internal cognitive state is a more robust and efficient termination signal than observing answer string consistency. Critically, Dynasor-CoT's reliance on generating a full candidate answer at each step to check for string consistency introduces substantial runtime overhead. In contrast, TRACE's lightweight probing of the model's belief state is far more computationally efficient.

### 5.3 ABLATION STUDY

To strictly validate the design rationale and robustness of TRACE, we conducted a comprehensive ablation study comprising a component-wise analysis and a hyperparameter sensitivity analysis. The component-wise evaluation detailed in Appendix C.1 demonstrates that removing either the belief stability check or the answer confidence check results in significant accuracy degradation. These findings confirm that the synergy between belief stability and answer confidence is critical

Table 2: **Main results on task performance and inference efficiency.** The Vanilla CoT baseline (highlighted in gray) serves as the primary performance reference. While aggressive early-exit methods suffer significant accuracy degradation in exchange for token savings, TRACE consistently achieves accuracy comparable or superior to Vanilla CoT with substantial token reduction, showcasing a more effective performance-efficiency trade-off. Among the efficiency-focused methods, the best and second-best performing results are marked in **bold** and underlined, respectively. "Acc." denotes Accuracy, and "Tok." denotes Average Generated Tokens. ↑ indicates higher is better; ↓ indicates lower is better.

| Method | NQ | | TriviaQA | | SQuAD | | HotpotQA | |
|---|---|---|---|---|---|---|---|---|
| | Acc. ↑ | Tok. ↓ | Acc. ↑ | Tok. ↓ | Acc. ↑ | Tok. ↓ | Acc. ↑ | Tok. ↓ |
| *Qwen3-8B* | | | | | | | | |
| Vanilla CoT | 68.95 | 1130 | 87.72 | 892 | 68.59 | 1123 | 55.17 | 1342 |
| NoThinking | 63.17 | 246 | 85.75 | 211 | 61.78 | 282 | 49.94 | 218 |
| Dynasor-CoT | 66.33 | 556 | 87.58 | 638 | 66.52 | 882 | 53.85 | 758 |
| DEER | 63.52 | 274 | 84.37 | 249 | 65.45 | 421 | 51.57 | 303 |
| TRACE (Ours) | **68.52** | 694 | **88.24** | 468 | **68.70** | 605 | **54.93** | 613 |
| *Qwen3-14B* | | | | | | | | |
| Vanilla CoT | 68.89 | 836 | 87.93 | 761 | 69.02 | 909 | 56.52 | 1089 |
| NoThinking | 63.13 | 258 | 84.80 | 256 | 63.87 | 282 | 51.69 | 329 |
| Dynasor-CoT | 67.32 | 456 | 87.08 | 683 | 68.37 | 726 | 54.32 | 765 |
| DEER | 65.17 | 366 | 85.26 | 282 | 64.13 | 317 | 51.80 | 443 |
| TRACE (Ours) | **68.85** | 503 | **87.64** | 485 | **68.58** | 551 | **55.73** | 626 |
| *DeepSeek-R1-Distill-Qwen-7B* | | | | | | | | |
| Vanilla CoT | 62.19 | 743 | 83.70 | 719 | 59.60 | 764 | 47.70 | 1005 |
| NoThinking | 56.46 | 337 | 78.18 | 313 | 52.51 | 460 | 41.67 | 368 |
| Dynasor-CoT | 62.25 | 706 | 83.23 | 685 | **59.58** | 758 | **47.62** | 853 |
| DEER | 58.20 | 444 | 79.53 | 437 | 54.57 | 520 | 43.28 | 494 |
| TRACE (Ours) | **62.83** | 575 | **83.52** | 518 | 59.32 | 675 | 47.20 | 668 |
| *DeepSeek-R1-Distill-Qwen-14B* | | | | | | | | |
| Vanilla CoT | 67.84 | 732 | 87.68 | 616 | 66.56 | 732 | 56.09 | 899 |
| NoThinking | 61.08 | 298 | 83.84 | 241 | 60.30 | 366 | 50.47 | 303 |
| Dynasor-CoT | **67.91** | 691 | 86.45 | 465 | 65.81 | 672 | 55.81 | 815 |
| DEER | 63.93 | 342 | 84.35 | 290 | 60.82 | 388 | 51.95 | 433 |
| TRACE (Ours) | 67.58 | 428 | **87.20** | 398 | **66.03** | 503 | **56.30** | 603 |
| *Llama3-8B* | | | | | | | | |
| Vanilla CoT | 66.82 | 1047 | 86.53 | 823 | 65.24 | 1012 | 53.49 | 1254 |
| NoThinking | 60.48 | 229 | 83.12 | 198 | 58.87 | 258 | 47.76 | 213 |
| Dynasor-CoT | 64.91 | 523 | 86.14 | 612 | 64.15 | 849 | 51.88 | 721 |
| DEER | 61.79 | 264 | 83.92 | 241 | 62.53 | 398 | 49.51 | 292 |
| TRACE (Ours) | **66.53** | 651 | **87.19** | 453 | **65.77** | 579 | **53.24** | 592 |

for achieving the optimal trade-off between efficiency and accuracy. Furthermore, the sensitivity analysis presented in Appendix C.2 reveals that TRACE exhibits a broad robustness plateau. The framework maintains high performance and efficiency across a wide range of parameter settings around our recommended safe thresholds, proving that it functions effectively in diverse RAG scenarios without requiring hyperspecific tuning.

## 6 CONCLUSION

In this paper, we address the overthinking challenge in RAG by identifying a unique **Exploratory-Synthesizing** paradigm that undermines conventional efficiency methods. We introduce **TRACE**, a novel, training-free framework that actively intervenes in the reasoning process. TRACE effectively curtails redundant reasoning by introspecting on the model's cognitive state, first ensuring sufficient knowledge exploration via belief state stability and then confirming high confidence in a synthesized conclusion. Our extensive experiments demonstrate that this approach significantly improves inference efficiency by reducing token generation while maintaining or improving task performance.

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

## A  CASE STUDY

To provide a concrete illustration of the "Exploratory-Synthesizing" pattern and the failure mode of conventional termination heuristics, we present a case study from the HotpotQA dataset, as detailed in Figure 4. The task is to identify the founding year of Virginia Commonwealth University (VCU). The retrieved documents, however, primarily discuss the university's basketball program, creating a potential for distraction.

This example clearly demonstrates the phenomenon of premature confidence. During the exploration stage, the model processes Document 1 and encounters a statement that the 2011-12 season was the "44th seaso" of the men's basketball program. Based on this single piece of evidence, a simple calculation (201244=1968) is performed. A standard confidence-based method like DEER registers a high confidence score (0.924) for the incorrect answer "1968" and prematurely terminates the reasoning process. This decision conflates the local confidence derived from a simple extraction and calculation with the global confidence required for a synthesized, correct answer. The model incorrectly assumes the founding year of the basketball program is the same as the founding year of the university.

In contrast, our TRACE framework successfully navigates this challenge. Although an initial high-confidence signal is detected, the belief state has not yet stabilized, failing the first condition of our dual-check mechanism. TRACE therefore continues the reasoning process, entering the synthesis stage. It integrates additional knowledge, recognizing that VCU was formed in 1968 through the merger of two institutions with earlier origins. This deeper synthesis leads to the correct answer,"1838". At this point, the model's belief state has stabilized (KL divergence = 0.007) and its confidence in the correct answer is high (0.949). With both conditions of the dual-check mechanism satisfied, TRACE terminates, having reached an accurate conclusion efficiently.

This case study empirically validates our central claim: in complex RAG scenarios, effective reasoning requires distinguishing between the exploration of evidence and the synthesis of a final conclusion. TRACE's dual-check mechanism proves robust against the misleading signals of premature confidence that cause simpler heuristics to fail.

## B  CORRELATION ANALYSIS OF BELIEF STABILITY, ANSWER CONFIDENCE, AND CORRECTNESS

The TRACE framework incorporates a dual-check mechanism that utilizes belief stability and answer confidence to circumvent the premature confidence trap inherent in RAG scenarios. This appendix presents a fine-grained empirical analysis to validate the intrinsic relationships among belief stability, answer confidence, and final answer correctness throughout the reasoning process.

**Experimental Design**  The experiment involved 200 randomly sampled instances from the NQ and HotpotQA validation sets. We performed inference across four target models including Qwen3-8B, Qwen3-14B, DS-7B, and DS-14B. To capture the dynamic evolution of the cognitive states of the models, we employed interventionist probing at every reasoning step $t$. We recorded a three-dimensional tuple consisting of belief instability $\Delta_{KL}^{(t)}$, answer confidence $C(W_t)$, and answer correctness. These data points were subsequently mapped onto heatmaps. In these plots, the X-axis represents belief instability as calculated by the minimum KL divergence between the current belief

**Question: The 2011–12 VCU Rams men's basketball team, led by third year head coach Shaka Smart, represented Virginia Commonwealth University which was founded in what year?**

**Answer: 1838**

**Retrieval Documents：**
1. 2011–12 VCU Rams men's basketball team The 2011–12 VCU Rams men's basketball team represented Virginia Commonwealth University during the 2011–12 NCAA Division I men's basketball season. It was ... <\omitted>
2. 2010–11 VCU Rams men's basketball team The 2010–11 VCU Rams men's basketball team represented Virginia Commonwealth University in the Colonial Athletic Association conference during the 2010–11 NCAA ... <\omitted>
...
5. 2009–10 VCU Rams men's basketball team The 2009–10 VCU Rams men's basketball team represented Virginia Commonwealth University during the 2009–10 college basketball season. This was head coach Shaka Smart's first season at VCU. The Rams compete in the Colonial Athletic Association and played their home games at Stuart C. Siegel Center. They finished the season 27–9, 11–7 in CAA play and ... <\omitted>

Okay, so I need to figure out when Virginia Commonwealth University (VCU) was founded. The question is about the 2011–12 VCU Rams men's basketball team, led by Shaka Smart, and asks when VCU was founded.

First, I'll look through the provided documents to see if any of them mention the founding year of VCU. Let me go through each document.

Document 1: Talks about the 2011–12 team, their coach Shaka Smart, and their performance. It mentions it's the 44th season of the men's basketball program. So, if the 2011–12 season was the 44th, I can subtract 44 from 2012 to estimate the founding year. 2012 minus 44 is 1968. **Wait,** that might not be accurate because the first season might not be the same as the founding year of the university.

| Conf=0.924 | **DEER EXIT!** Answer:\boxed{**1968**} | ❌ |

Document 2: Discusses the 2010–11 team. It says it was Shaka Smart's second year and they played in the CAA. No mention of the university's founding year here.

Document 3: 2012–13 team, their move to the A10 conference. Again, no founding year mentioned.

Document 4: This seems to be a duplicate or similar to Document 1, as it starts the same way. It also mentions the 44th season, so same calculation as before: 2012 - 44 = 1968. But I'm not sure if the basketball program's founding is the same as the university's founding.

Document 5: 2009–10 team, Shaka Smart's first year. No founding year info here either.

So, none of the documents directly state when VCU was founded. However, Document 1 and 4 mention that the 2011–12 season was the 44th season of the men's basketball program. If I take 2012 and subtract 44, that gives me 1968. But I should verify if the basketball program started in 1968 or if the university itself was founded earlier.

Wait, I know that VCU is a relatively young university. It was established in 1838 as the Richmond Male Asylum, which later became the Richmond Professional Institute (RPI) in 1900, and then merged with the Medical College of Virginia (MCV) in 1968 to form VCU. So, the university as we know it was founded in 1838.

| KL div=0.007 & Conf=0.949 | **TRACE EXIT!** Answer:\boxed{**1838**} | ✓ |

But the basketball program might have started earlier. However, since the documents mention the 44th season in 2012, that would imply the program started in 1968 (2012 - 44 = 1968). But I'm a bit confused because VCU's founding as a combined university was in 1968, but the individual components had earlier origins.

Wait, the question is about when VCU was founded, not the basketball program. So, the university itself was founded in 1968 when RPI and MCV merged. Therefore, the answer should be 1968.\n</think>

The 2011–12 VCU Rams men's basketball team represented Virginia Commonwealth University, which was founded in 1968.

**Answer:** \boxed{1968}

| **Vanilla CoT** Answer:\boxed{**1968**} | ❌ |

Figure 4: **A case study from the HotpotQA dataset illustrating the failure of premature termination and the effectiveness of TRACE.** The task is to find the founding year of Virginia Commonwealth University (VCU). After an initial extraction, the model expresses hesitation with "Wait,". At this point, DEER triggers its check and, misled by the high confidence (0.924) in the spurious answer "1968", incorrectly terminates. In contrast, TRACE performs a more fine-grained check after each reasoning segment. It correctly determines that the reasoning is incomplete and continues until the belief state stabilizes (KL div=0.007) and the final conclusion is highly confident (conf=0.949), yielding the correct answer "1838". The standard Vanilla CoT also reaches the wrong conclusion due to reasoning errors after an unnecessarily long thought process.

distribution $D_t$ and all historical distributions $\{D_0, ..., D_{t-1}\}$. This metric follows the definition in Section 4.3 where lower values indicate higher stability. The Y-axis represents answer confidence derived from the geometric mean of the token probabilities for the elicited immediate answer. The color intensity of each grid cell corresponds to the average correctness rate of answers falling within that region, ranging from deep purple for incorrect answers to bright yellow for correct answers.

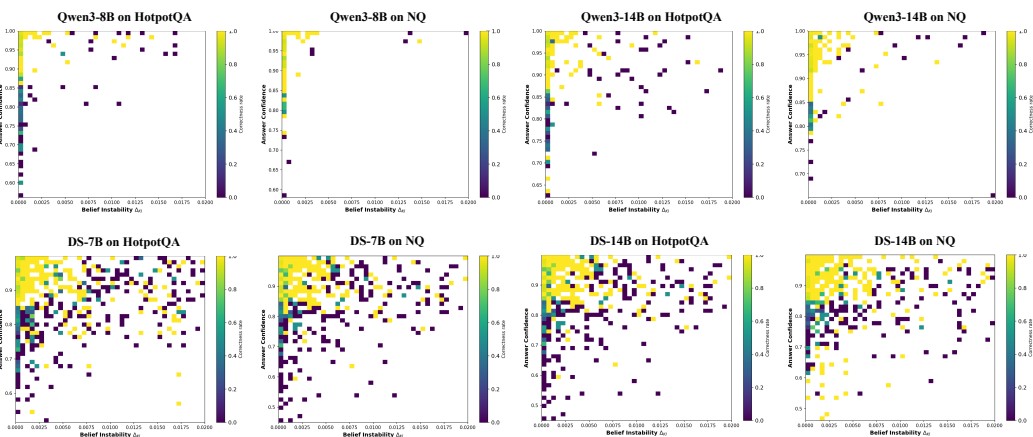

Figure 5: **Heatmap analysis of the correlation between belief instability, answer confidence, and correctness.** The analysis covers four models across the HotpotQA and NQ datasets. The X-axis represents Belief Instability $\Delta_{KL}$ and the Y-axis represents Answer Confidence $C(W_t)$. The color intensity indicates the correctness rate where yellow approximates 1.0 and purple approximates 0.0. DeepSeek-R1-Distill-Qwen models are abbreviated as DS. The results demonstrate that answer correctness is significantly higher when both high belief stability and high answer confidence are satisfied, which corresponds to the top-left region.

**Results and Analysis** Figure 5 visualizes the distribution of correctness across different cognitive states and reveals several critical conclusions.

First, a prominent feature across all plots is the presence of deep purple regions in the upper-right and central areas which indicate near-zero correctness. These regions correspond to states characterized by high answer confidence where $C > 0.9$ but high belief instability where $\Delta_{KL} > 0.0075$. This phenomenon empirically verifies the existence of the premature confidence trap. It shows that models frequently generate confident yet incorrect answers during unstable exploration phases. This observation explains the failure of confidence-only baselines like DEER in RAG settings.

Second, we observe that stability alone is insufficient to guarantee correctness when focusing on the far-left sector of the X-axis where $\Delta_{KL} \approx 0$. The grid cells in the middle-to-lower sections of this region often remain dark or exhibit mixed colors. This indicates that a model may lack confidence in the answer even when it achieves a stable belief state. This finding justifies the necessity of introducing confidence as a second validation check.

Finally, the highest correctness rates are densely and exclusively concentrated in the top-left corner of the heatmaps. This region represents the intersection of high belief stability where $\Delta_{KL}$ approaches 0 and high answer confidence where $C$ approaches 1.0. This result provides compelling evidence that the TRACE dual-check mechanism precisely targets this optimal exit condition. It effectively filters out both premature errors and low-confidence stable states.

While the macroscopic distribution patterns are universal, significant microscopic differences exist between model families. The Qwen3 models exhibit a smoother and continuous transition in the heatmaps where the high-correctness golden region forms a solid and cohesive block. In contrast, the DeepSeek-Distill models display a more discrete and scattered pattern. The high-correctness region appears slightly fragmented and is accompanied by more noise in lower-confidence areas. We hypothesize that this may be a result of the distillation process. Student models can mimic the final output accuracy of teacher models but their internal belief state calibration may be less refined compared to base pre-trained models. Despite these distributional variations, the dual-criteria

strategy of TRACE remains robust as the top-left corner consistently remains the region of maximal correctness across both model families.

## C COMPREHENSIVE ABLATION STUDIES

This appendix presents a detailed breakdown of the ablation studies summarized in Section 5.3. We divide the analysis into two parts: a component-wise evaluation to verify the necessity of the dual-check mechanism and a hyperparameter sensitivity analysis to assess the framework's robustness.

### C.1 COMPONENT-WISE ABLATION ANALYSIS

To isolate the contribution of each condition in TRACE's dual-check mechanism, we evaluated two degraded variants on the HotpotQA dataset using Qwen3 models. The first variant, **w/o Exploration Check**, removes the condition $\Delta_{KL}^{(t)} \leq \theta_{exp}$. This effectively reduces the framework to a standard confidence-based early exit strategy, terminating generation as soon as $C(W_t) > \theta_{inf}$. The second variant, **w/o Confidence Check**, removes the condition $C(W_t) > \theta_{inf}$ and terminates reasoning solely based on the stabilization of the belief state ($\Delta_{KL}^{(t)} \leq \theta_{exp}$).

The quantitative results are summarized in Table 3. The **w/o Exploration Check** variant achieves the lowest token consumption but suffers from a substantial drop in accuracy (e.g., Qwen3-8B drops from 54.93% to 51.57%). This empirically confirms that relying solely on confidence metrics in RAG scenarios leads to premature exits triggered by partial evidence, validating the necessity of the belief stability check to enforce sufficient exploration. The **w/o Confidence Check** variant also exhibits suboptimal performance. While it ensures the model's belief has converged, the lack of a confidence verification allows the model to terminate on answers that are stable but incorrect (e.g., stabilizing on a "I don't know" state or a hallucination). In contrast, the full **TRACE** framework achieves the highest accuracy, demonstrating that both the exploration check and the confidence check are indispensable. The exploration check prevents "under-thinking," while the confidence check ensures the quality of the final synthesis.

Table 3: **Component-wise Ablation Study on HotpotQA.** We compare the full TRACE framework against variants lacking either the Exploration Check ($\Delta_{KL}$) or the Confidence Check ($C(W)$). The results show that removing either component leads to significant accuracy degradation, confirming their mutual necessity.

| Model | Method Variant | Accuracy ↑ | Avg. Tokens ↓ |
|---|---|---|---|
| Qwen3-8B | TRACE (Full) | **54.93** | 613 |
| | - w/o Exploration Check | 51.57 | **303** |
| | - w/o Confidence Check | 51.63 | 448 |
| Qwen3-14B | TRACE (Full) | **55.73** | 626 |
| | - w/o Exploration Check | 51.80 | **443** |
| | - w/o Confidence Check | 52.11 | 519 |

### C.2 HYPERPARAMETER SENSITIVITY ANALYSIS

To validate the stability and robustness of the TRACE framework under different configurations, we conducted a detailed hyperparameter sensitivity experiment on the HotpotQA dataset. This section aims to quantitatively analyze how variations in the belief stability threshold ($\theta_{exp}$) and the answer confidence threshold ($\theta_{inf}$) dynamically affect the model's reasoning accuracy and computational efficiency, thereby establishing optimal operational intervals and revealing the underlying synergistic mechanisms.

To provide a robust and efficient starting point without the need for exhaustive grid searches, we selected $\theta_{exp} = 0.0075$ and $\theta_{inf} = 0.90$ as the **safe initialization thresholds** for this analysis based on the heatmap observations in the previous section. This selection is empirically grounded: the heatmaps indicate that $\theta_{inf} = 0.90$ serves as a critical safety boundary, below which accuracy

drops precipitously, thus constituting a necessary baseline for quality assurance. Meanwhile, the stability threshold of $\theta_{exp} = 0.0075$ lies at the edge of the high-accuracy plateau, minimizing invalid redundant reasoning while covering the vast majority of correct instances. Based on this baseline, we fixed one parameter while varying the other to observe the behavioral boundaries of the models across dimensions of belief stability and answer confidence.

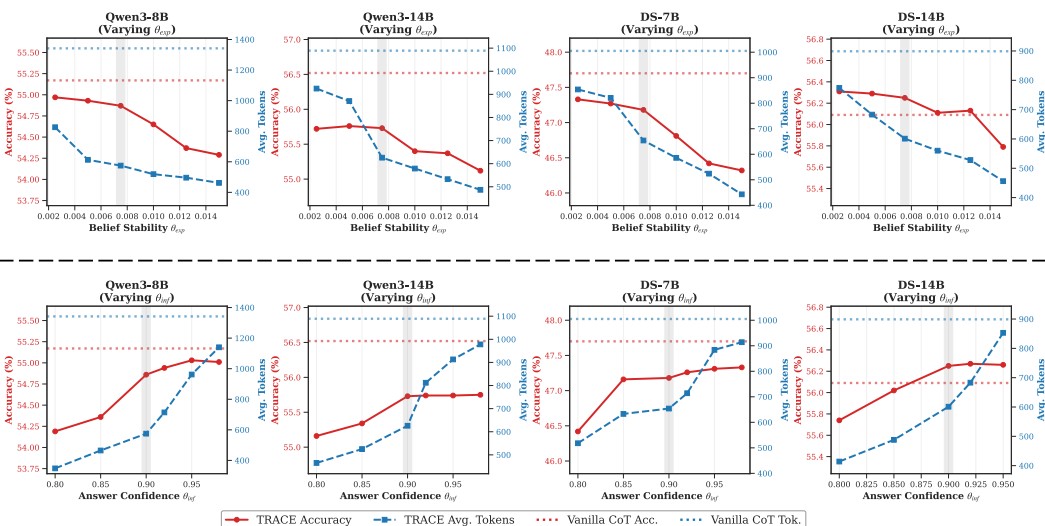

Figure 6: **Hyperparameter sensitivity analysis on HotpotQA.** The top row illustrates the impact of varying the belief stability threshold $\theta_{exp}$ (with $\theta_{inf}$ fixed at 0.90), while the bottom row shows the impact of varying the answer confidence threshold $\theta_{inf}$ (with $\theta_{exp}$ fixed at 0.0075). The red solid lines represent Accuracy, and the blue dashed lines represent Average Tokens. The vertical gray bars highlight the selected safe thresholds. DeepSeek-R1-Distill-Qwen models are abbreviated as DS. The results demonstrate robust performance plateaus around these safe points.

Regarding the impact of the belief stability threshold $\theta_{exp}$, we fixed $\theta_{inf} = 0.90$ and observed variations in $\theta_{exp}$ from 0.0025 to 0.0150. As illustrated in the top row of Figure 6, when $\theta_{exp}$ is within the strict interval of 0.0025 to 0.0075, the accuracy curves for all models exhibit a significant plateau with minimal fluctuation. This indicates that further tightening the threshold within this range yields negligible performance gains. However, the corresponding token consumption increases linearly with the tightening of the threshold; for instance, the token count for Qwen3-8B surges from 575 to 926. This implies that an overly strict stability requirement introduces substantial computational redundancy. Conversely, when the threshold is relaxed beyond 0.0075, although token consumption decreases further, accuracy begins to show a non-negligible decline, particularly in the DeepSeek series models, which exhibit a more pronounced downward trend. This phenomenon verifies that 0.0075 is an ideal equilibrium point balancing efficiency and accuracy, avoiding both overthinking and premature exit before belief convergence.

Regarding the impact of the answer confidence threshold $\theta_{inf}$, we fixed $\theta_{exp} = 0.0075$ and observed variations in $\theta_{inf}$ from 0.80 to 0.98. The experimental results in the bottom row of Figure 6 clearly reveal the role of $\theta_{inf}$ as a "safety gate." When the threshold is lowered from 0.90 to 0.80, the accuracy for all models drops sharply, directly confirming that the low-confidence interval is replete with plausible-sounding but incorrect hallucinations that must be strictly filtered. On the other hand, as the threshold is raised from 0.90 to 0.98, while accuracy remains stable, token consumption climbs steeply, approaching the original levels of Vanilla CoT. This suggests that pursuing extreme confidence often entails diminishing marginal returns and does not contribute to final outcome improvement. Therefore, $\theta_{inf} = 0.90$ is identified as the critical watershed for ensuring model output reliability, effectively delineating the boundary between error risk and computational cost.

# D    SENSITIVITY ANALYSIS ON THE NUMBER OF RETRIEVED DOCUMENTS

This section investigates how varying the number of retrieved documents ($k$) influences model reasoning dynamics and validates the robustness of the TRACE framework under varying degrees of information density and noise. Retrieval-Augmented Generation (RAG) systems face a fundamental trade-off: increasing $k$ provides richer context but simultaneously introduces more noise and irrelevant information, often leading models into redundant "overthinking." We conducted experiments on the HotpotQA dataset with $k \in \{1, 3, 5, 10\}$, representing scenarios ranging from information scarcity to information overload. We compared the performance of Vanilla CoT, DEER, and TRACE (configured with the safe thresholds $\theta_{exp} = 0.0075$, $\theta_{inf} = 0.90$) across the Qwen3-8B and Qwen3-14B models. The detailed results are presented in Table 4.

Table 4: **Impact of Retrieved Document Count ($k$) on Performance and Efficiency.** We compare Vanilla CoT, DEER, and TRACE across varying numbers of retrieved documents ($k \in \{1, 3, 5, 10\}$) on the HotpotQA dataset. TRACE maintains accuracy comparable to Vanilla CoT while significantly reducing token consumption, especially in high-$k$ scenarios where information redundancy is high.

| Model | Documents | Method | Accuracy (%) | Avg. Tokens |
|---|---|---|---|---|
| **Qwen3-8B** | $k = 1$ | Vanilla CoT | 46.51 | 1082 |
| | | DEER | 41.49 | 256 |
| | | **TRACE** | 45.45 | 522 |
| | $k = 3$ | Vanilla CoT | 53.14 | 1219 |
| | | DEER | 47.40 | 270 |
| | | **TRACE** | 52.81 | 593 |
| | $k = 5$ | Vanilla CoT | 55.17 | 1342 |
| | | DEER | 51.57 | 303 |
| | | **TRACE** | 54.93 | 613 |
| | $k = 10$ | Vanilla CoT | 55.03 | 1533 |
| | | DEER | 51.52 | 362 |
| | | **TRACE** | 54.91 | 705 |
| **Qwen3-14B** | $k = 1$ | Vanilla CoT | 48.11 | 889 |
| | | DEER | 41.91 | 386 |
| | | **TRACE** | 47.51 | 525 |
| | $k = 3$ | Vanilla CoT | 54.49 | 941 |
| | | DEER | 49.53 | 401 |
| | | **TRACE** | 53.65 | 589 |
| | $k = 5$ | Vanilla CoT | 56.52 | 1089 |
| | | DEER | 51.80 | 443 |
| | | **TRACE** | 55.73 | 626 |
| | $k = 10$ | Vanilla CoT | 56.32 | 1215 |
| | | DEER | 51.71 | 478 |
| | | **TRACE** | 55.68 | 665 |

The experimental results highlight distinct failure modes for the baseline methods. Vanilla CoT suffers from an efficiency bottleneck: as $k$ increases from 1 to 10, the average token consumption for Qwen3-8B surges from 1082 to 1533. This confirms that without intervention, models tend to indiscriminately process all provided context, causing computational costs to grow linearly with information load. Conversely, DEER exhibits a persistent accuracy deficit. Even as more information becomes available (increasing $k$ from 1 to 10), DEER fails to close the performance gap with Vanilla CoT. For instance, at $k = 10$, DEER's accuracy on Qwen3-8B plateaus at 51.52%, significantly lower than Vanilla CoT's 55.03%. This limitation stems from DEER's sole reliance on a confidence threshold, leading to *premature exit*: the model halts generation upon encountering the first piece of information (often a distractor or partial evidence common in high-$k$ settings) that triggers high local confidence. Increasing $k$ merely adds more potential distractors early in the context, preventing the model from reaching the correct, comprehensive evidence located later in the sequence.

In contrast, TRACE successfully addresses both issues through its dual-check mechanism. By incorporating a belief stability check ($\theta_{exp}$), TRACE ensures that the model does not exit based on a transient spike in confidence caused by partial evidence. Instead, it forces the reasoning process to continue until the internal belief state stabilizes, implying that sufficient evidence has been synthesized. As a result, in the high-noise $k = 10$ scenario, TRACE achieves an accuracy of 54.91%—virtually identical to Vanilla CoT (55.03%) and superior to DEER—while capping token consumption at 705, representing a **54%** reduction in computational cost. This demonstrates TRACE's unique ability to filter out noise and terminate reasoning exactly when information sufficiency is reached, regardless of the context length.

## E   RUNTIME EFFICIENCY ANALYSIS

To evaluate the practical efficiency of our proposed TRACE framework, we conducted a series of runtime performance tests comparing it against the Vanilla CoT baseline. The experiments were performed on a server equipped with four NVIDIA RTX 5090 GPUs. We utilized the vLLm framework for optimized inference deployment. For each test, we first warmed up the system with 50 samples to ensure stable performance, and then measured the total execution time required to process a subsequent batch of 500 samples. We varied the number of retrieved documents, denoted by $k$, using values of 1, 3, and 5 to simulate different context loads.

Table 5: **Execution time costs (in seconds) for Vanilla CoT and TRACE with varying numbers of retrieved documents** ($k$)**.** The speedup values are computed as the time reduction of TRACE over Vanilla CoT. For layout purposes, DeepSeek-R1-Distill-Qwen models are abbreviated as DS.

| Documents | Method | Time Cost (s) | | | |
|---|---|---|---|---|---|
| | | Qwen3-8B | Qwen3-14B | DS-7B | DS-14B |
| $k = 1$ | Vanilla CoT | 101.34 | 105.52 | 62.14 | 113.27 |
| | TRACE | 84.48 | 91.61 | 50.58 | 82.96 |
| | **Speedup** | **×0.17** | **×0.13** | **×0.19** | **×0.27** |
| $k = 3$ | Vanilla CoT | 130.58 | 153.65 | 79.46 | 159.82 |
| | TRACE | 82.19 | 113.70 | 67.41 | 113.85 |
| | **Speedup** | **×0.37** | **×0.26** | **×0.15** | **×0.29** |
| $k = 5$ | Vanilla CoT | 163.95 | 182.30 | 91.88 | 193.52 |
| | TRACE | 79.52 | 127.11 | 76.08 | 133.97 |
| | **Speedup** | **×0.51** | **×0.30** | **×0.17** | **×0.31** |

The results of our efficiency comparison are detailed in Table 5. The data unequivocally demonstrates that TRACE provides a significant and robust improvement in runtime efficiency over the standard CoT approach across all tested models and configurations. First, for every model and every value of $k$, the execution time for TRACE is consistently lower, confirming that its mechanism of curtailing redundant reasoning steps translates directly into tangible time savings in a real-world deployment scenario. The Speedup metric, which represents the percentage of time saved by TRACE, is substantial across the board, ranging from a notable 13% (for Qwen3-14B at $k = 1$) to a remarkable 51% (for Qwen3-8B at $k = 5$). This level of performance gain is critical for practical applications, as it directly corresponds to lower computational costs and reduced user-perceived latency. We observe that TRACE's efficiency advantage becomes more pronounced as the context becomes more complex (i.e., as $k$ increases). For instance, with the Qwen3-8B model, the time saved grows from 17% at $k = 1$ to 37% at $k = 3$, and further to an impressive 51% at $k = 5$. This trend strongly suggests that as the input context becomes richer, the propensity for Vanilla CoT to overthink increases significantly. In contrast, TRACE's introspection mechanism excels in these document-heavy scenarios by accurately identifying the point where reasoning is sufficient, thereby halting the generation of unnecessary tokens and yielding progressively greater time savings. In conclusion, this empirical analysis validates that TRACE is a highly effective framework for improving the computational efficiency of LRLMs in RAG settings.

