# OpenReview forum: "TRACE: Adaptive Curtailment of Reasoning in Retrieval-Augmented Generation via Trajectory Reflection"
_ICLR.cc/2026/Conference — ICLR 2026 Conference Desk Rejected Submission_

### Official Review · Reviewer_gtP5 · 2025-10-28

**Soundness:** 3
**Presentation:** 3
**Contribution:** 2
**Rating:** 6
**Confidence:** 3

**Summary:**

This manuscript investigates how to decrease the thought length of large reasoning models regarding for the RAG setting. This manuscript first demonstrated the failure of existing methods and provided the reasons. Then, this manuscript proposes a new method with new metrics to address the problems in RAG settings. The experiments on various models and various benchmarks showed the effectiveness of the proposed methods.

**Strengths:**

1. Propose an effective and convenient methods to address the overthinking problems of reasoning models.
2. The writing is good and the presentation is clear, which make this manuscript is easy to understand.
3. The results show that the proposed methods is competitive with baselines.

**Weaknesses:**

1. I have some concerns about the experiment setting. First, Qwen 3 8B and Qwen3 14B are not reasoning models. Second, the selected RAG tasks only require about 1k outputs, which cannot represent the long outputs of reasoning models[1].

2. Regarding the results, providing the latency of the reasoning thought can directly demonstrate the speed-up ratio. I am curious about the comparison with [1].


[1] R-KV: Redundancy-aware KV Cache Compression for Reasoning Models

**Questions:**

1. What's the motivation for compressing the reasoning thoughts of LRM over the RAG setting? What's the redundancy？

2. Is this method compatible with the situation without RAG?

---

> ### Author Response · Authors · 2025-11-20
> **Response to Reviewer gtP5**
>
> We sincerely thank the reviewer for the suggestions, particularly the feedback regarding the experimental setup and efficiency analysis. We address your specific concerns as follows.
>
> **Response to Weakness 1.** Regarding the classification of "reasoning models," **our setting aligns with the prevailing conventions in efficient reasoning literature** (consistent with the baselines we compared, such as DEER and Dynasor-CoT). In this context, LRLMs are characterized as models capable of executing complex, multi-step CoT processes to solve tasks, rather than being limited to specialized architectures. Both Qwen3 and the newly added Llama-3 fully justifying their inclusion as reasoning models. Regarding output length, although the average output for RAG tasks is approximately 1k tokens, this constitutes a **significant latency bottleneck** in real-time interaction scenarios. The core value of TRACE lies in **saving hundreds of unnecessary reasoning tokens** while maintaining accuracy. While the mechanism of TRACE is applicable to scenarios with longer outputs, the 1k token length is sufficient to verify the efficiency gains achieved by eliminating overthinking.
>
> **Response to Weakness 2.** Regarding the latency metrics, we have added a comprehensive Runtime Efficiency Analysis in **Appendix E (Table 5)**. The results show that TRACE achieves significant end-to-end speedups in real-world vLLM deployments. Regarding the comparison with R-KV, we first clarify the fundamental difference in **optimization scope**. TRACE addresses **cognitive redundancy** at the semantic level by algorithmically determining when to halt generation to reduce the total token count, whereas R-KV addresses **representational redundancy** at the system level by compressing the KV cache to lower memory usage and accelerate attention computation. We acknowledge R-KV as a significant contribution to system efficiency. Importantly, we consider these two approaches to be **orthogonal and highly complementary**. While we could not conduct a joint integration experiment in this revision due to **computational resource constraints**, we believe that combining TRACE's token pruning with R-KV's cache compression represents a promising direction for maximizing future inference efficiency.
>
> **Response to Question 1.** The necessity of compressing reasoning thoughts in Large Reasoning Language Models originates from the **overthinking phenomenon** prevalent in general non-RAG scenarios. Models often generate substantial repetitive confirmations or invalid reasoning steps even after obtaining a definite answer. This computation failing to provide additional information increments constitutes redundancy. In RAG scenarios, this issue persists and becomes **more severe**. With the introduction of external documents, the model faces a more complex information environment and is prone to endless exploration of irrelevant document fragments or repeated comparison of known information. This leads to an **explosive growth of redundant reasoning**. The design of TRACE specifically targets this amplified redundancy by **monitoring belief stability** to precisely identify the point of information saturation and thereby effectively cutting off invalid exploration paths.
>
> **Response to Question 2.** TRACE is **fully compatible** with non-RAG scenarios. To demonstrate this, we evaluated TRACE by **Qwen3-8B** and **Qwen3-14B** against Vanilla CoT and DEER on NQ and TriviaQA datasets without retrieving external documents. As shown in the table below, TRACE successfully maintains accuracy levels comparable to Vanilla CoT across both models and datasets. We observe that TRACE is slightly less aggressive in token pruning compared to DEER in this specific setting. This is because TRACE employs a stricter criterion for reasoning completeness, requiring both **belief stability** and **answer confidence** to be satisfied, whereas DEER triggers an exit based solely on confidence. Despite this more conservative approach, TRACE still achieves meaningful efficiency gains of approximately **12-15\%** while providing a robust safeguard for accuracy. We emphasize that the core advantage of TRACE is maximized in **RAG scenarios**, where the high redundancy of reasoning allows TRACE to achieve significantly larger reduction ratios.
>
> **Table: Performance Comparison in Non-RAG Settings**
>
> | Dataset | Model | Method | Accuracy (%) | Avg. Tokens |
> | :--- | :--- | :--- | :---: | :---: |
> | **NQ** | **Qwen3-8B** | Vanilla CoT | 46.71 | 483 |
> | | | DEER | 46.47 | 352 |
> | | | **TRACE** | **46.63** | **414** |
> | | **Qwen3-14B** | Vanilla CoT | 51.46 | 421 |
> | | | DEER | 51.29 | 305 |
> | | | **TRACE** | **51.25** | **368** |
> | **TriviaQA** | **Qwen3-8B** | Vanilla CoT | 78.65 | 612 |
> | | | DEER | 78.29 | 447 |
> | | | **TRACE** | **78.48** | **536** |
> | | **Qwen3-14B** | Vanilla CoT | 84.43 | 658 |
> | | | DEER | 84.55 | 482 |
> | | | **TRACE** | **84.38** | **573** |

---

> > ### Comment · Reviewer_gtP5 · 2025-11-26
> >
> > Thank you for the detailed response; most of my concerns have been addressed.  In general, I am pleased to maintain my positive score to support this paper.
> >
> > However, I do not agree that TRACE and RKV are orthogonal and highly complementary. Both methods are based on the same “overthinking” idea and differ mainly in whether they compress the tokens or the KV cache.

---

> ### Author Response · Authors · 2025-12-02
>
> We sincerely thank Reviewer gtP5 for the positive feedback and continued support of our paper. We are pleased that the revised version has addressed most of your concerns.
>
> We agree with your observation that both methods are based on the same “overthinking” idea.
>
> Our previous description of them as orthogonal and complementary was intended to emphasize their differences at the implementation. TRACE focuses on preventing LRLMs from generating potentially redundant tokens, while R-KV focuses on compressing the KV cache of generated tokens.

---

### Official Review · Reviewer_mt9Z · 2025-11-01

**Soundness:** 3
**Presentation:** 3
**Contribution:** 3
**Rating:** 6
**Confidence:** 4

**Summary:**

This paper addresses an efficiency challenge in Large Reasoning Language Models (LRLMs) operating within RAG frameworks. The authors identify a fundamental limitation of existing confidence-based early-exit methods (e.g., DEER) in RAG settings: they are misled by premature confidence, where models exhibit high certainty after processing only partial evidence, leading to incorrect answers and degraded accuracy. Through empirical analysis, the authors characterize an Exploratory-Synthesizing pattern in RAG reasoning, contrasting it with the convergent deduction pattern of non-RAG scenarios. To mitigate this, they propose TRACE, a training-free framework that employs a cascading, dual-check mechanism. At each reasoning step, TRACE first assesses the stability of the model's predictive belief state to ensure sufficient knowledge exploration and then verifies high confidence in a synthesized final answer before termination. Extensive experiments demonstrate that TRACE significantly reduces token generation while maintaining accuracy compared to standard Chain-of-Thought prompting.

**Strengths:**

(1) The motivation of the paper is clear. The paper reveals that RAG introduces a unique reasoning dynamic that systematically undermines existing efficiency methods. The identification and quantitative validation of the "premature confidence" phenomenon contributes to the area and the idea of "Exploratory-Synthesizing" is reasonable.

(2) The proposed TRACE framework is solid. Its training-free nature ensures broad applicability without the cost of additional fine-tuning. The dual-check mechanism is well-justified.

(3) The experimental design is thorough, employing multiple models and diverse benchmarks to establish generalizability.  The ablation study is effective.

**Weaknesses:**

(1)  The framework relies on two key thresholds, θ_{exp} and θ_{inf} , which are tuned per dataset. While a grid search protocol is described, the paper does not deeply explore the sensitivity of the results to these values. The generalizability of these optimal thresholds across different model families or task domains should be discussed.

(2) Only some selected QA datasets are used for evaluation. There are a large number of datasets available for RAG evaluation, and a wider range of evaluation should be conducted. Please check https://github.com/RUC-NLPIR/FlashRAG for more datasets available.

**Questions:**

(1) Could the authors provide more insight into the failure modes of the belief stability check? Are there scenarios where the belief state stabilizes on an incorrect hypothesis, and how might TRACE be extended to detect and correct such cases?

(2) Given that the belief state  is derived from a single token, how does the method perform with answers that are phrases or entities whose first token might be ambiguous or uninformative? Has the team experimented with multi-token probing strategies?

(3) How will TRACE be integrated into more advanced RAG paradigms, such as those involving iterative retrieval or self-critique (e.g., Self-RAG)? Could the trajectory reflection module be used to trigger not just termination, but also a retrieval of additional documents if exploration is deemed insufficient?

---

> ### Author Response · Authors · 2025-11-20
> **Response to Reviewer mt9Z**
>
> We sincerely thank the reviewer for the positive assessment of our work, particularly the recognition of the Exploratory-Synthesizing pattern and the rationale behind the TRACE framework design. We address your specific questions and suggestions as follows.
>
> **Response to Weakness 1.** We conducted a detailed hyperparameter sensitivity analysis in **Appendix C.2** to assess the stability of TRACE under different configurations. The experimental results demonstrate that TRACE exhibits a **broad performance plateau**. As long as the hyperparameters remain within the reasonable exploration range recommended in the main text, both inference accuracy and token pruning efficiency show high stability. This strongly evidences that TRACE possesses **high robustness to parameter variations** and **does not require specific fine-tuning** for different model families to function effectively.
>
> **Response to Weakness 2.** We selected NQ, TriviaQA, SQuAD, and HotpotQA for evaluation because they cover a **diverse range of task types**, from single-hop factual retrieval to complex multi-hop reasoning, ensuring **broad representativeness**. Current experimental results show that TRACE achieves **consistent performance improvements** across these tasks of varying difficulty. Although constrained by computational resources, we were unable to include all datasets from FlashRAG in this experiment, but we agree that evaluation on a wider range of benchmarks is valuable and plan to further expand our testing scope in future work.
>
> **Response to Question 1.** We explored the potential failure modes of the belief stability check in the heatmap analysis within **Appendix B**. The heatmaps reveal that within the region of extremely low belief instability where $\Delta_{KL} \approx 0$, there are still some samples with low confidence, represented by the **dark area in the bottom-left corner**. This corresponds to the scenario you mentioned where the model might **stabilize on an incorrect or uncertain hypothesis**. This is precisely why we designed the **dual-check mechanism**; relying solely on belief stability is insufficient, and the **answer confidence check** must be combined as a second line of defense. When the belief is stable but confidence remains below the threshold, TRACE identifies this as an invalid stability state. Our current strategy is to allow the model to continue reasoning, but in future extensions, we consider using this signal to trigger error correction mechanisms, guiding the model to actively retrieve higher-quality evidence.
>
> **Response to Question 2.** We adopt a probing strategy based on the first token primarily due to two core considerations. The first is the **balance between computational efficiency and empirical validity**. This method maximizes the utilization of the existing KV Cache, enabling real-time monitoring with **almost no additional computational overhead**. Although a single token may have semantic ambiguity for certain long entities, our correlation analysis in **Appendix B** confirms that the changes in its probability distribution are **statistically significant** enough to accurately characterize the convergence trend of the model's overall belief. The second is the **robustness of the evaluation metric**. Our goal is to quantify the stability of the internal cognitive state formed by the model based on the current reasoning history, rather than predicting the stability of the specific generated content. Introducing multi-token probing would actually introduce **lexical selection noise** during the decoding process, such as the model fluctuating between phrases that are semantically identical but phrased differently. This would cause **spurious fluctuations** in the sensitive KL divergence metric, thereby interfering with the judgment of the true belief state.
>
> **Response to Question 3.** The modular design of TRACE makes it highly suitable for integration into advanced paradigms such as Self-RAG or iterative retrieval. In particular, our dual-check mechanism can provide **precise trigger signals** for iterative retrieval. For instance, when the model passes the belief stability check but fails the answer confidence check, this clearly indicates a **state of information scarcity**. In advanced RAG systems, this specific state can be used to **trigger a new round of document retrieval** or initiate a self-reflection module, rather than simply continuing generation or terminating. This is also a direction we are currently exploring.

---

### Official Review · Reviewer_fBbi · 2025-11-01

**Soundness:** 3
**Presentation:** 2
**Contribution:** 2
**Rating:** 2
**Confidence:** 3

**Summary:**

This paper proposes TRACE, a training-free framework designed to prevent premature early exit during reasoning in RAG framework.
The method aims to adaptively curtail reasoning by comparing belief-state distributions across reasoning steps, measured through KL divergence and combining this signal with an answer-confidence check. The authors argue that this dual criterion helps distinguish between two reasoning phases: Exploration and Synthesis, mitigating the issue of premature termination caused by overconfident early retrieval. Experiments on several QA datasets (NQ, TriviaQA, SQuAD, HotpotQA) using Qwen and DeepSeek models suggest that TRACE reduces reasoning length (22 ~ 54% token savings) without accuracy loss relative to Chain-of-Thought (CoT) prompting or DEER-style early exit.

**Strengths:**

1. $\textbf{Clear motivation and problem framing.}$
The paper identifies an important limitation in RAG applied with CoT prompting: confidence-based early exit methods often fail when external documents cause inflated confidence before synthesis is complete. This diagnosis is intuitively appealing and practically relevant.
2. $\textbf{Training-free and easily deployable method.}$
TRACE can be applied to any LLM pipeline without retraining or model modification. Its compatibility with existing RAG setups makes it attractive from an engineering perspective.
3. $\textbf{Empirical efficiency improvements.}$
The reported token reduction (22 ~ 54%) and runtime gains are impressive, especially given that accuracy is maintained on most benchmarks.

**Weaknesses:**

1. $\textbf{Unconvincing causal link between distributional stability and early-exit mitigation.}$
The central claim that comparing belief distributions over reasoning steps can prevent premature exits is not sufficiently supported.
As shown in Figure 1, the “with-document” setting maintains high confidence from the start, yet it is unclear how distributional comparison (e.g., small KL divergence) can reliably distinguish premature local certainty from true synthesis completion.
Stabilized distributions might simply indicate consistent but incomplete retrieval reasoning rather than successful synthesis. The paper provides no direct analysis linking KL stability to epistemic completeness or correctness.
2. $\textbf{Limited experimental depth and analysis.}$z
Despite mentioning “extensive experiments,” the evaluation primarily includes only two result tables (Tables 2–3).
There is no detailed analysis of the relationship between distributional dynamics and performance, nor ablation on threshold sensitivity or task type. Visualization or correlation studies between KL divergence and correctness would be essential to substantiate the paper’s theoretical motivation.
3. $\textbf{Hyperparameter and baseline configuration unclear.}$
The paper briefly mentions empirical choices of thresholds ($\theta_\text{exp}, \theta_\text{inf}$) in Appendix A.1 but does not describe details of performance and length trend when the hypereparameters were different.
Similarly, baseline methods such as DEER or Self-RAG are presented without clarification of whether comparable hyperparameter tuning was performed. This omission makes the fairness of the comparison somewhat uncertain.
4. $\textbf{No analysis on the number of retrieved documents.}$
Section 4.1 notes that the experiments use the top-5 retrieved passages, but there is no study on how performance or early-exit stability changes with different document counts (e.g., top-3 vs. top-10). Given that premature confidence is tied to retrieval context size, such an ablation is crucial.

**Questions:**

1. In Figure 1, confidence remains high from the first to the last step in the “with document” setting. How can TRACE detect such premature confidence if belief distributions are already stable early?
2. How sensitive is TRACE to the thresholds ($\theta_\text{exp}, \theta_\text{inf}$)? Were there any specific pattern between the thresholds and performance or generation length?
3. Were similar tuning efforts (e.g., grid search or threshold sweeping) performed for baseline methods such as DEER or Dynasor-CoT?
4. The paper mentions using top-5 retrieved passages in RAG. How does performance vary when the number of retrieved documents changes (e.g., top-3 vs. top-10)? Is the method robust to such variations compared to baselines?
5. Have the authors verified whether KL stability correlates with semantic completeness, such as evidence coverage or synthesis correctness?

---

> ### Author Response · Authors · 2025-11-20
> **Response to Reviewer fBbi**
>
> We appreciate the reviewer's insightful comments and address the specific concerns as follows.
>
> **Response to Question 1: Operational logic regarding premature confidence.**
> We clarify that TRACE avoids premature confidence not by detecting the phenomenon itself, but by enforcing a **cascading verification mechanism**. In TRACE, the **belief stability check** serves as a strict prerequisite. Even if the model exhibits high confidence during the early stages, TRACE prevents termination as long as the internal belief distribution is still shifting. Simultaneously, even if the belief stabilizes early, reasoning continues as long as the confidence has not reached our set threshold.
>
> **Response to Weakness 1, 2 and Questions 5: Validity of belief stability check.**
> We conducted a fine-grained correlation analysis in **Appendix B** to validate our dual-check mechanism. The heatmaps in **Figure 5** clearly show that high correctness rates are **mostly** concentrated in the region where both high belief stability and high answer confidence are satisfied (**the top-left yellow region**). States satisfying only high confidence ($C>0.9$) or high stability often contain numerous failed samples (purple regions). This empirically confirms that due to the premature confidence trap, relying solely on high confidence or high stability is unreliable. It also proves that belief stability is a necessary indicator for distinguishing true synthesis completion from premature local certainty.
>
> **Response to Weakness 3 and Question 2: Hyperparameter sensitivity.**
> We performed a detailed sensitivity analysis in **Appendix C.2**. **Figure 6** shows that TRACE exhibits a broad **performance plateau**. As long as parameters are within a reasonable range (the exploration range provided in the main text), accuracy remains stable near the baseline level while token consumption is significantly reduced. This confirms that TRACE is highly robust to parameter variations and **does not require hyper-specific tuning** to function effectively across different models. Using general "safe thresholds" yields performance close to SOTA without exhaustive search.
>
> **Response to Question 3: Fairness of baseline comparison.**
> We strictly adopted the **optimal hyperparameter configurations recommended in the original literatures** for all baselines. We argue that further grid search for baselines yields negligible gains for two reasons:
> 1.  **Inherent Robustness:** As shown in Appendix C.2, TRACE achieves optimal performance even with general thresholds, indicating our advantage stems from the methodology, not overfitting.
> 2.  **Structural Limitation:** Our heatmap analysis (**Appendix B**) reveals that high-confidence incorrect answers are inextricably mixed with correct ones in the confidence dimension. Thus, **no single confidence threshold**—even if perfectly tuned—can effectively separate them. TRACE outperforms baselines because its **dual-check mechanism** resolves this fundamental indistinguishability.
>
> **Response to Weakness 4 and Question 4: Sensitivity to retrieved document count ($k$).**
> We added a sensitivity analysis varying $k \in \{1, 3, 5, 10\}$ in **Appendix D**. As shown in **Table 4** (excerpt below), **TRACE** maintains accuracy comparable to Vanilla CoT across all settings and achieves **greater efficiency gains** as information redundancy increases (e.g., **54% token reduction** at $k=10$). This demonstrates high robustness to retrieval context variations.
>
> **Table: Impact of Retrieved Document Count ($k$) on Performance and Efficiency (HotpotQA)**
>
> | Model | Documents | Method | Accuracy (%) | Avg. Tokens |
> | :--- | :---: | :--- | :---: | :---: |
> | **Qwen3-8B** | $k=1$ | Vanilla CoT | 46.51 | 1082 |
> | | | DEER | 41.49 | 256 |
> | | | **TRACE** | 45.45 | 522 |
> | | $k=3$ | Vanilla CoT | 53.14 | 1219 |
> | | | DEER | 47.40 | 270 |
> | | | **TRACE** | 52.81 | 593 |
> | | $k=5$ | Vanilla CoT | 55.17 | 1342 |
> | | | DEER | 51.57 | 303 |
> | | | **TRACE** | 54.93 | 613 |
> | | $k=10$ | Vanilla CoT | 55.03 | 1533 |
> | | | DEER | 51.52 | 362 |
> | | | **TRACE** | **54.91** | **705** |
> | **Qwen3-14B** | $k=1$ | Vanilla CoT | 48.11 | 889 |
> | | | DEER | 41.91 | 386 |
> | | | **TRACE** | 47.51 | 525 |
> | | $k=3$ | Vanilla CoT | 54.49 | 941 |
> | | | DEER | 49.53 | 401 |
> | | | **TRACE** | 53.65 | 589 |
> | | $k=5$ | Vanilla CoT | 56.52 | 1089 |
> | | | DEER | 51.80 | 443 |
> | | | **TRACE** | 55.73 | 626 |
> | | $k=10$ | Vanilla CoT | 56.32 | 1215 |
> | | | DEER | 51.71 | 478 |
> | | | **TRACE** | **55.68** | **665** |

---

> > ### Comment · Reviewer_fBbi · 2025-11-27
> >
> > After reading the authors’ response and the new analyses (Appendix B–D), I think my main concerns have been largely addressed:
> > - The correlation heatmaps now provide evidence that the dual-check (belief stability + confidence) is empirically linked to correctness.
> > - The threshold sensitivity study shows a reasonably wide plateau, which reduces my worry about fragile hyperparameter tuning.
> > - The experiments varying the number of retrieved documents suggest that TRACE is robust to retrieval context size, while maintaining efficiency gains over CoT and DEER.
> >
> > I still recommend surfacing some of these key analyses into the main paper for clarity, but overall I now find the method sufficiently supported and practically useful.
> >
> > I am updating my rating to 6 .

---

> ### Author Response · Authors · 2025-12-02
>
> We sincerely thank Reviewer fBbi for the detailed and valuable comments, which greatly benefited our rebuttal and revision of the paper. We are grateful for your decision to raise the score.
>
> We agree with your suggestion to incorporate the key analyses to the main text and will implement this change in the final version.

---

### Official Review · Reviewer_nASG · 2025-11-06

**Soundness:** 3
**Presentation:** 3
**Contribution:** 2
**Rating:** 6
**Confidence:** 3

**Summary:**

This paper addresses a crucial problem of overthinking in large reasoning models under RAG scenarios. Most of the existing frameworks are based on the assumption that reasoning depends only on the model's internal, parametric knowledge.

**Strengths:**

Strengths:
1. This paper addresses an unaddressed problem of overthinking in LLMs under RAG conditions.
2. TRACE is a novel method that addresses the overthinking under RAG.

**Weaknesses:**

Weaknesses:
1. Mosty Qwen family is considered for the experiments.

**Questions:**

1. Is the same phenomenon seen for other models such as Llama or open-source models?
2. Is there a scaling law?

---

> ### Author Response · Authors · 2025-11-20
> **Response to Reviewer nASG**
>
> We sincerely thank the reviewer for the constructive feedback and address the specific questions as follows.
>
> **Response to Question 1.** To verify the universality of the premature confidence phenomenon, we extended our experiments to include the **Llama-3-8B** model. The results confirm our hypothesis:
>
> **1. Confirmation of Premature Confidence (Data from Table 1):**
> As shown in the table below, Llama-3-8B exhibits a consistent "premature confidence" pattern across all datasets in **RAG settings**. The confidence-based baseline (DEER) suffers significant accuracy drops (e.g., **-4.60%** on HotpotQA) compared to Vanilla CoT, indicating that the model is misled by high local confidence during exploration.
>
> | **Llama-3-8B Performance** | **NQ** | **TriviaQA** | **SQuAD** | **HotpotQA** |
> | :--- | :---: | :---: | :---: | :---: |
> | **Vanilla CoT Acc. (%)** | 68.34 | 87.14 | 67.79 | 57.38 |
> | **DEER Acc. (%)** | 63.04 | 82.91 | 63.55 | 52.78 |
> | **Accuracy Drop** | **-5.30** | **-4.23** | **-4.24** | **-4.60** |
>
> **2. Effectiveness of TRACE (Data from Table 2):**
> TRACE successfully mitigates this issue. As shown below, TRACE restores accuracy to levels comparable to (or even better than) Vanilla CoT while achieving substantial token reductions (e.g., **~52.8%** reduction on HotpotQA).
>
> | **Llama-3-8B Main Results** | **NQ** (Acc / Tok) | **TriviaQA** (Acc / Tok) | **SQuAD** (Acc / Tok) | **HotpotQA** (Acc / Tok) |
> | :--- | :---: | :---: | :---: | :---: |
> | **Vanilla CoT** | 66.82 / 1047 | 86.53 / 823 | 65.24 / 1012 | 53.49 / 1254 |
> | **DEER** | 61.79 / 264 | 83.92 / 241 | 62.53 / 398 | 49.51 / 292 |
> | **TRACE (Ours)** | **66.53 / 651** | **87.19 / 453** | **65.77 / 579** | **53.24 / 592** |
>
> These findings confirm that TRACE is a robust, model-agnostic solution.
>
> **Response to Question 2.** Regarding the scaling laws, we compared the performance data of 7B/8B models against 14B models as detailed in Table 2 and Appendix E. Our observations indicate that smaller models tend to gain relatively larger efficiency improvements from the active pruning of TRACE. However, based on current observations, we prefer to treat the derivation of definite scaling laws with caution. This is because inference efficiency and the magnitude of token pruning in RAG scenarios do not depend solely on model parameter size. They are also significantly influenced by the retrieval context such as document quantity and quality. As we demonstrated in the sensitivity analysis in Appendix D, changing the number of retrieved documents alters both the reasoning dynamics and the pruning ratio. Therefore, model size is not the only determining factor as the complexity of the retrieval environment is also a key variable affecting the overthinking phenomenon.

---

### Author Response · Authors · 2025-11-24
**Summary of Revisions**

We sincerely thank all reviewers for their constructive and insightful comments. We are greatly encouraged that the reviewers found our problem definition **"crucial"**, our method **"novel"** and **"solid"**, and our presentation **"clear"**.

In this revision, we have extensively updated the manuscript to address your concerns, adding **Three new appendix sections** and **multiple comprehensive experiments**. The revised manuscript has expanded by 6 pages compared to the initial submission. The key updates are summarized as follows:

1.  **Expanded Model Scope:** We incorporated **Llama3-8B** into all main experiments (**Table 1** & **Table 2**) to verify the architectural generalizability of our findings.
2.  **Mechanism Validation:** We added **Appendix B (Figure 5)** featuring fine-grained correlation heatmaps. This empirically validates the existence of the "premature confidence" trap and confirms that both belief stability and answer confidence are necessary indicators for measuring the reasoning state.
3.  **Robustness Analysis:** We introduced **Appendix C.2** for a detailed hyperparameter sensitivity analysis and **Appendix D** for a sensitivity analysis regarding the number of retrieved documents. These analyses demonstrate the stability of TRACE across diverse settings.
4.  **Manuscript Refinement:** We performed a thorough proofreading and revision based on the valuable suggestions from all reviewers, aiming to better highlight the uniqueness and broad impact of our work.

Below, we address the specific comments from each reviewer point-by-point. We deeply appreciate the reviewers' guidance in improving our paper and remain fully available to answer any further questions. We look forward to your feedback.

---

### Note · Program_Chairs · 2026-01-17
**Submission Desk Rejected by Program Chairs**

The following references in this submission do not refer to real documents and/or have major errors in bibliographic information:

 Wenjie Ma, Chuhuai Yue, Anqi Zhang, Yihe Zhang, and Yi Zhou. NoThinking: A preliminary study on reasoning models without thinking. arXiv preprint arXiv:2405.14283, 2024.